# Pairwise Elastic Net Representation-Based Classification for Hyperspectral Image Classification

**DOI:** 10.3390/e23080956

**Published:** 2021-07-26

**Authors:** Hao Li, Yuanshu Zhang, Yong Ma, Xiaoguang Mei, Shan Zeng, Yaqin Li

**Affiliations:** 1School of Mathematics and Computer Science, Wuhan Polytechnic University, Wuhan 430023, China; lihao@whpu.edu.cn (H.L.); zengshan1981@whpu.edu.cn (S.Z.); leeyaqin@whpu.edu.cn (Y.L.); 2Electronic Information School, Wuhan University, Wuhan 430072, China; 2018202120089@whu.edu.cn (Y.Z.); meixiaoguang@whu.edu.cn (X.M.)

**Keywords:** hyperspectral image (HSI) classification, sparse representation, collaborative representation, pairwise elastic net, neighbor information

## Abstract

The representation-based algorithm has raised a great interest in hyperspectral image (HSI) classification. $l_{1}$-minimization-based sparse representation (SR) attempts to select a few atoms and cannot fully reflect within-class information, while $l_{2}$-minimization-based collaborative representation (CR) tries to use all of the atoms leading to mixed-class information. Considering the above problems, we propose the pairwise elastic net representation-based classification (PENRC) method. PENRC combines the $l_{1}$-norm and $l_{2}$-norm penalties and introduces a new penalty term, including a similar matrix between dictionary atoms. This similar matrix enables the automatic grouping selection of highly correlated data to estimate more robust weight coefficients for better classification performance. To reduce computation cost and further improve classification accuracy, we use part of the atoms as a local adaptive dictionary rather than the entire training atoms. Furthermore, we consider the neighbor information of each pixel and propose a joint pairwise elastic net representation-based classification (J-PENRC) method. Experimental results on chosen hyperspectral data sets confirm that our proposed algorithms outperform the other state-of-the-art algorithms.

## 1. Introduction

A hyperspectral image is a 3D remote sensing image containing hundreds of bands, from visible to infrared spectra. Due to their abundant spectral information, HSIs have become an actual application in the field of remote sensing, such as skin imaging [1], ground elements identifying [2] and mineral exploration [3]. To date, many classification algorithms for hyperspectral datasets have been proposed. Among the techniques, the support vector machine (SVM)  [4], Gaussian mixture-model (GMM) [5] and the Gaussian maximum-likelihood classifier (MLC) [6] are all proved to be effective for solving HSI classification problem. The most concerning research methods in recent years can be roughly divided into two categories: representation-based algorithms and deep learning-based algorithms. On the one hand, in order to make full use of the spectral and spatial information of HSIs, some effective spectral–spatial feature extraction methods have been combined with sparse models to improve the characterization capability of models, such as  [7,8,9,10]. On the other hand, since the deep convolutional neural network (CNN) with deep architecture has been proven to be very effective in using image features, this type of method using deep CNN for hyperspectral feature extraction has stimulated various studies [11,12,13,14].

This paper is mainly focused on the HSI classification algorithm based on representation learning. The classification principle of the method is to assume that each testing pixel can be reconstructed with labeled training pixels. Then, the abundance coefficients of the testing pixel can be obtained with the penalty of $l_{1}$-norm or $l_{2}$-norm, which is named sparse representation classification (SRC) [15] and collaborative representation classification (CRC) [16]. In [17], Chen et al.  first introduced the sparsity model into hyperspectral classification and proposed the joint sparse representation classification (JSRC) method by incorporating the contextual information. In [18], considering that different atoms have different importance for the reconstruction process, Li et al.  proposed the nearest regularized subspace (NRS) classifier with Tikhonov regularization. By wisely combing SRC and KNN, in Ref. [19] a class-dependent sparse representation classifier (cdSRC) was proposed. However, some research [18,20] shows that the collaboration of approximation enhances classification results rather than competition. Therefore, in Ref. [21], a joint within-class CRC was provided to solve the HSI classification tasks. In [22], the kernel version of CRC was further considered and the Kernel-based CRC (KCRC) was proposed. There are also some investigations dedicated to improving classification effectiveness. On the one hand, some focus on the more simple and robust dictionary to reduce computation costs. On the other hand, some take the neighborhood spatial information as an important factor in improving classification accuracy. In Ref. [23], the nonlocal joint collaborative representation (NJCRC) algorithm was proposed by utilizing a subdictionary whose atoms are obtained by the k-nearest neighbor (K-NN) with testing samples rather than the whole dictionary atoms. In [24], Fang et al.  introduced the shape irregular neighbor region into the joint SRC model and proposed the shape adaptive joint sparse representation (SAJSRC).

It is worth noting that both the SRC-based algorithms and CRC-based algorithms have their limitations. In these representation-based classification models, the obtained abundance coefficients reflect the importance of each training sample for reconstruction. Accordingly, the primary concern of this type of method is the solution of the abundance coefficient. Ideally, the test pixels should be linearly represented by atoms from the same category. The nonzero terms of sparse coefficients should be located at the position of the corresponding class. For SRC, it tends to select as few atoms as possible. The too sparse property will lead to the deviation of the absolute reconstruction error, and the sparsity will be weakened when the number of training atoms sets is small. For CRC, it tends to select all the atoms for reconstruction, and the class discrimination will be weak when including mixed-class information. Intuitively, SRC and CRC should be balanced to achieve better classification performance is necessary.

To solve the above problem, in Ref. [25], the elastic net representation-based classification (ENRC) method was proposed. The elastic net originally raised in [26] encourages both sparsity and grouping by forming a convex combination of the CRC and SRC governed by a selectable parameter. Furthermore, the elastic net can yield a sparse estimate with more than n nonzero weights. Based on these advantages, the ENRC improves of HSI classification performance. However, the optimal balance factors are all obtained by traversing the manufactured parameter space. This makes the algorithm time-consuming and complex. Additionally, the pixelwise fusion algorithm cannot make full use of the spatial information of the HSI.

Fortunately, the recent literature [27] has pointed out that the pairwise elastic net (PEN) model using similarity measures between regressors can establish a local balance between SRC and CRC. It can achieve more flexible grouping than ENRC. Moreover, PEN allows the customization of the sparsity relationship between any two features. Hence, in this work, we propose the pairwise elastic net representation-based classification (PENRC) method to overcome the indigenous disadvantages of ENRC, SRC and CRC. It can automatically achieve the balance between $l_{1}$-norm and $l_{2}$-norm so that more robust weight coefficients can be estimated, and further realizing better between-class sparse and intraclass collaborative classification performance.

Specifically, the main contribution of the proposed PENRC can be briefly summarized as follows. First, considering the computation cost when using all the dictionaries, we adopt the KNN to select the labeled atoms, which are more similar to the testing pixel as an optimal sub-dictionary. Then, unlike the ENRC, which assigns only a single global tradeoff between sparsity and collaboration, we introduce a similar matrix about sub-dictionary atoms in penalties, resulting in the local sparsity and collaboration tradeoff and be more flexible than ENRC. After obtaining the abundance coefficients, we use the principle of minimum reconstruction error to decide the final label. We also provide a further extension of our algorithm by incorporating the neighbor information of each pixel.

In summary, it is expected that the abundance coefficients from PENRC reveal a more powerful discriminant ability, thereby outperforming the original SRC, CRC and ENRC.

The remaining parts of the paper are organized as follows: Section 2 briefly introduces the two classical SRC and CRC classifiers. Section 3 details the proposed PENRC mechanism. Section 4 gives the experimental results on chosen two datasets. Finally, Section 5 concludes this paper.

## 2. Related Works

Denoting a testing pixel as $y=\left[y_{1},\ldots ,y_{B}\right]\in R^{B\times 1}$ and the dictionary composed of training atoms with class order as $X=\left[X_{1},\cdots ,X_{C}\right]\in R^{B\times N}$, where *B* is the number of spectral bands, $N=\sum_{c=1}^{C}N_{c}$ is the training atoms number and *C* is the total number of categories. The sub-dictionary $X_{c}\in R^{B\times N_{c}}$ is the set of training atoms in *c*-th class.

### 2.1. Sparse Representation for HSI Classification

The sparse model assumes that a testing pixel can be linearly approximated with few dictionary atoms suitably [15]. Then, for a testing pixel y, the purpose of SRC model is to obtain the corresponding abundance coefficients by minimizing the reconstruction error ${∥y-X\alpha^{SRC}∥}_{2}^{2}$ with the sparse constraint term ${∥\alpha^{SRC}∥}_{1}$. Mathematically, the object function can be represent as follows:(1)$${\hat{\alpha }}^{SRC}=\arg\min{∥y-X\alpha^{SRC}∥}_{2}^{2}+\lambda_{1}{∥\alpha^{SRC}∥}_{1},$$
where $\lambda_{1}$ is the balancing parameter. The weight vector $\alpha^{SRC}\in R^{{}^{N\times 1}}$ is sparse and only have few nonzero terms. It can be obtained by solving Equation (Equation 1) with basis pursuit (BP) or basis pursuit denoising (BPDN) algorithms [28,29]. When $l_{2}$-norm is directly used, Equation (Equation 1) can be solved by subspace pursuit (SP) and orthogonal matching pursuit (OMP) algorithms  [30].

After obtaining the weight vector $\alpha^{SRC}$, we can assign the final class label which corresponding the mimimum reconstruction error to the testing pixel:(2)$$\begin{array}{cc} class(y) & =\arg\min_{c=1,\cdots ,C}{∥y-\hat{y_{c}}∥}_{2}^{2}\\ \; & =\arg\min_{c=1,\cdots ,C}{∥y-X_{c}{\hat{\alpha }}_{c}^{SRC}∥}_{2}^{2}, \end{array}$$
where $\alpha_{c}^{SRC}$ is the subset of sparse vector $\alpha^{SRC}$ which belongs to *c*-th class.

### 2.2. Collaborative Representation for HSI Classification

Unlike SRC model, the CRC assumes that a testing pixel can be linearly combined with all the training set [21]. The CRC attempts to obtain abundance coefficients by minimizing the reconstruction error ${∥y-X\alpha^{CRC}∥}_{2}^{2}$ with the term ${∥\alpha^{CRC}∥}_{2}$. Thus, the CRC can be expressed as:(3)$${\hat{\alpha }}^{CRC}=\arg\min{∥y-X\alpha^{CRC}∥}_{2}^{2}+\lambda_{2}{∥\alpha^{CRC}∥}_{2},$$
where $\lambda_{2}$ balances the influence of the reconstruction error and constraint term. Equation (Equation 3) can be simply solved with a closed form. Assuming that the derivative of the above cost function and is zero, we can obtain the optimal value of $\alpha^{CRC}$:(4)$$\alpha^{CRC}={\left(X^{T}X+\lambda_{2}I\right)}^{-1}X^{T}y,$$
where I is an identity matrix with the size of N×N. After obtaining $\alpha^{CRC}$, the final class label *c* of testing pixel can be determined with the minimm residual rule as introduced in last section.

For the above representation-based classification methods, training atoms tend to be “competitive” in SRC due to the sparse constraints. With $l_{2}$-norm, all atoms participate in the representation process equally. Thus, CRC tends to be “cooperative”. Researchers compared the performance of SRC with CRC in literature [21,22]. Moreover, the experiments showed that in some cases, SRC performances better than CRC while CRC performance was better in other cases. For example, in remote sensing images, the SRC algorithm gave rise to a more remarkable improvement with some mixed pixels [31]. Thus, it is an effective way to combine SRC and CRC appropriately. In fact, in Ref. [25], FRC and ENRC algorithms to combine SRC with CRC were proposed. However, the dictionary chosen in [25] consists of all the training samples and brings a large computational burden. In addition, the algorithms in [25] only set a global trade-off between SRC and CRC, leading to the inflexible balance of different classes.

## 3. Proposed PENRC

The framework of our proposed PENRC algorithm is shown in Algorithm 1. First, we built a local adaptive dictionary to reduce the amount of calculation. Given a test pixel, we used the KNN algorithm to select the *K* pixels that are most similar to the local adaptive dictionary set. Second, we constructed the PENRC model of the hyperspectral image. We used the local adaptive dictionary to construct the PEN model and obtain the abundance coefficients corresponding to the testing pixel. Then, we calculated the reconstruction error of each class according to the abundance coefficients and used the minimum reconstruction error to classify the testing pixels. In addition, in order to further improve the classification performance, we also integrated the spatial information of the pixel neighborhood into the model, named joint pairwise elastic net representation-based classification (J-PENRC).
**Algorithm 1** the Proposed PEN Algorithm**Input:**      (1) $X\in R^{B\times N}$, the training set.                 (2) *K*, λ.**Procedure:***Step 1:* Obtain adaptive dictionary D by applying KNN.*Step 2:* Obtain weight vector $\hat{\alpha }$ according to Equation (Equation 8):                  for i=1:N                  update ${\hat{\alpha }}_{i}$ by Equations (Equation 19) and  (Equation 20).*Step 3:* Decide the final label $\mathrm{class}\left(y\right)$ by the minimum reconstruction error principle by Equation (Equation 14).**Output:**       $\mathrm{class}\left(y\right)$.

### 3.1. Local Adaptive Dictionary

In representation-based methods, dictionaries are usually composed of all labeled training pixels [32,33]. In order to have a robust representation, it is necessary to ensure that the dictionary is complete (that is, enough training samples are needed). However, training samples are usually limited in practice. In addition, using all training pixels directly will lead to a large amount of computation. Therefore, to solve the above problems, we utilize the local adaptive dictionary to obtain a more robust representation.

For a testing pixel y, we utilize the KNN to construct a similar signal set D as the adaptive dictionary. However, due to the high dimension of the hyperspectral image, it is unreasonable to directly use Euclidean distance to measure the similarity of the spectral vector. In order to increase the separability of data, LDA [34] algorithm is used to project HSIs into low-dimensional space, which can find an optimal projection direction to minimize the intraclass distance of samples and maximize the inter-class distance. Let $\Gamma \in R^{B{}^{\prime }\times B}$ indicate the LDA mapping matrix and $B{}^{\prime }$ represent the reduced dimension. Then, the similarity measure between the testing atom y and arbitrary training atoms $x_{n}$ can be expressed as:(5)$$d_{n}=∥\Gamma y-\Gamma x_{n}∥.$$
Then, we sorted all the distance set $\left[x_{1},x_{2},\cdots ,x_{N}\right]$ in descending order and obtained the dictionary indices ${\left\{i_{c}\right\}}_{c=1,\cdots C}$ corresponding to the first *K* large distance values. The adaptive dictionary can be denoted as:(6)$$D=X\left(:,i_{c}\right),c=1,\cdots ,C.$$

### 3.2. Pairwise Elastic Net Representation Based Classification

First, we introduce the concept of correlation matrix. Consider the following two matrices $R_{1}$ and $R_{2}$:(7)$$\begin{array}{cc} R_{1}=\left(\begin{array}{ccc} 1.0 & 0.5 & 0.5\\ 0.5 & 1.0 & 0.5\\ 0.5 & 0.5 & 1.0 \end{array}\right) & R_{2}=\left(\begin{array}{ccc} 1.0 & 0.9 & 0.0\\ 0.9 & 1.0 & 0.3\\ 0.0 & 0.3 & 1.0 \end{array}\right) \end{array}.$$

We can see that the three features in the $R_{1}$ matrix have the same similarity values. At this point, it is effective to set the global trade-off between $l_{1}$-norm and $l_{2}$-norm. Nevertheless, for the matrix $R_{2}$, feature 1 is very similar to feature 2 (regarding $l_{2}$-norm), feature 1 is independent from feature 3 (regarding $l_{1}$-norm) and feature 2 is slightly related to feature 3 (regarding elastic net). Hence, we need a flexible trade-off scheme to match the regularization term with the data structure.

Thus, the objective function of our proposed PENRC can be denoted as:(8)$$\hat{\alpha }=\arg\min{∥y-D\alpha ∥}_{2}^{2}+\lambda \left({∥\alpha ∥}_{2}^{2}+{∥\alpha ∥}_{1}^{2}-{\left|\alpha \right|}^{T}R\left|\alpha \right|\right),$$
where R is the similarity matrix between atoms in the adaptive dictionary $D\in R^{K\times K}$. Some frequently-used similarity measures are absolute atom correlation $R_{ij}=\left|D_{i}^{T}D_{j}\right|$ and Gaussian kernel $R_{ij}=\exp\left(-{∥D_{i}-D_{j}∥}^{2}/\sigma^{2}\right)$ et al. Considering some basic results and notation with abundance coefficients and similarity matrix:(9)$$\begin{array}{cc} \; & {∥\alpha ∥}_{2}^{2}={\left|\alpha \right|}^{T}I\left|\alpha \right| \end{array}$$(10)$$\begin{array}{cc} \; & {∥\alpha ∥}_{1}={\left|\alpha \right|}^{T}1=1^{T}\left|\alpha \right| \end{array}$$(11)$$\begin{array}{cc} \; & {∥\alpha ∥}_{1}^{2}={\left|\alpha \right|}^{T}11^{T}\left|\alpha \right|, \end{array}$$
where I is the identity matrix and 1 is a vector of all ones. Then, the fourth term in Equation (Equation 8) representing the trade-off between $l_{1}$-norm and $l_{2}$-norm can be explained as follows. For the completely similar features, $R=11^{T}$. Equation (Equation 8) only left $l_{2}$-norm, reducing the impact of the $l_{1}$ constraint. For the completely dissimilar features, R=I, and Equation (Equation 8) reduces to SRC model with only $l_{1}$ constraint. That is to say, when the two features are similar, we take the CRC method; when the two features are dissimilar, we take the SRC method; for the remaining cases, we take the ENRC method. Thus, the flexible trade-off scheme can be realized though our proposed PENRC.

To further enhance the classification performance, we also incorporate the spatial information of HSI pixel into the PENRC model. In [24], a shape adaptive (SA) region is proposed for each pixel. In our work, we utilized the neighbor information with SA and the chosen pixel can be represented by the average of all pixels in the SA window. For an arbitrary pixel y in the HSI, the corresponding SA set matrix can be denoted as $Y_{SA}=\left[y_{1},y_{2},\cdots ,y_{T}\right]$. T is the number of chosen pixels in SA. Then, the pixel y introduced into spatial information can be obtained by
(12)$${\bar{y}}_{SA}=\frac{1}{T}\sum_{t=1}^{T}y_{T},$$

Then, the sparse coefficients $\alpha_{SA}$ for ${\bar{y}}_{SA}$ can be denoted as:(13)$$\begin{array}{cc} {\hat{\alpha }}_{SA} & =\arg\min∥{\bar{y}}_{SA}-{\bar{D}}_{SA}\alpha_{SA}{∥}_{2}^{2}\\ \; & +\lambda \left({∥\alpha_{SA}∥}_{2}^{2}+{∥\alpha_{SA}∥}_{1}^{2}-{\left|\alpha_{SA}\right|}^{T}R\left|\alpha_{SA}\right|\right), \end{array}$$

Once the sparse coefficients $\alpha_{SA}$ is obtained, the final label can be determined by the category minimum reconstruction error:(14)$$\mathrm{class}\left(y\right)=\arg\min_{c=1,\cdots ,C}{∥{\bar{y}}_{SA}-{\bar{D}}_{c_{SA}}\alpha_{c_{SA}}∥}_{2},$$
where,${\bar{D}}_{c_{SA}}$ and $\alpha_{c_{SA}}$ represent the subset of ${\bar{D}}_{SA}$ and $\alpha_{SA}$ corresponding to *c*-th class, respectively.

### 3.3. Coordinate Descent

To solve Equation (Equation 8), we rewrite it as following:(15)$$\hat{\alpha }=\arg\min{∥y-D\alpha ∥}_{2}^{2}+\lambda {\left|\alpha \right|}^{T}P\left|\alpha \right|,$$
where $P=I+11^{T}-R$. As [27] proves, only if P has nonnegative entries and is a positive semidefinite (PSD) matrix, the second term ${\left|\alpha \right|}^{T}P\left|\alpha \right|$ in above model is convex. However, the matrix P in Equation (Equation 15) is not always a PSD matrix. We can consider the following way as proved in [27]:(16)$$P_{\theta }^{S}=\theta I+\left(1-\theta \right)P,$$
where $\frac{\tau }{\tau +1}\leq \theta \leq 1$ and $\tau =-\min\left\{0,\lambda_{min}\left(P\right)\right\}$.

Then, Equation (Equation 15) can be seen as a quadratic program (QP) problem and can be solved by the QP solver. However, the QP solver does not meet the high-dimensional data requirements. In order to obtain more exact results, we use the coordinate descent method [35] in this paper. The approach can be summarized as: given a convex function $f\left(\alpha \right)$, we calculate the derivative $\frac{\partial f}{\partial \alpha_{i}}$; update $\alpha_{i}$ by holding all $\alpha_{j}$ (where j≠i) fixed with the equation $\frac{\partial f}{\partial \alpha_{i}}=0$; cyclic each $\alpha_{i}$ iteratively until the termination condition is satisfied.

In PENRC, we have
(17)$$\begin{array}{cc} f\left(\alpha \right) & ={\arg\min∥y-D\alpha ∥}_{2}^{2}+\lambda {\left|\alpha \right|}^{T}P\left|\alpha \right|\\ \; & =y^{T}y-2q^{T}\alpha +\alpha^{T}Q\alpha +\lambda \sum_{i,j}P_{i,j}\left|\alpha_{i}\alpha_{j}\right|, \end{array}$$
where P is PSD and nonnegative, $Q=D^{T}D$ and $q=X^{T}y$. Then, the derivative $\frac{\partial f}{\partial \alpha_{i}}$ is
(18)$$\frac{\partial f}{\partial \alpha_{i}}=-2q_{i}+2Q_{i}^{T}\alpha +2\lambda \mathrm{sgn}\left(\alpha_{i}\right)\sum_{j=1}^{K}P_{i,j}\left|\alpha_{j}\right|.$$

If the derivative is 0, we update $\alpha_{i}$ according to $\alpha_{-i}=\alpha_{\left\{1:K\right\}\backslash i}$:(19)$$\left(Q_{ii}+P_{ii}\right)\alpha_{i}+\mathrm{sgn}\left(\alpha_{i}\right)\lambda \sum_{j\neq i}P_{ij}\left|\alpha_{j}\right|=q_{i}-\sum_{j\neq i}Q_{ij}\alpha_{j}.$$

Then, we define the scalars a,b and *c*. Let $a=Q_{ii}+P_{ii}$, $b=\lambda \sum_{j\neq i}P_{ij}\left|\alpha_{j}\right|$ and $c=q_{i}-\sum_{j\neq i}Q_{ij}\alpha_{j}$. The update equation can be denoted as:(20)$$\alpha_{i}=\left\{\begin{array}{cc} \left(c+b\right)/a & c<-b\\ 0 & -b\leq c\leq b\\ \left(c-b\right)/a & c>b. \end{array}\right.$$

## 4. Results

In this part, to validate the superiority of our proposed PENRC, we compare our proposed PENRC (pixelwise) and PENRC with both the single pixel-based and spatial information-based algorithms such as the KNN [36], SRC [15], CRC [16], fused representation-based classification (FRC) method [25], elastic net representation-based classification (ENRC) method [25], nearest regularized subspace (NRS) classifier [18], shape adaptive joint sparse representation classification (SAJSRC) [24] and weighted joint nearest neighbor and sparse representation (WINN-JSR) [37]. All the experiments are conducted using MATLAB R2014b on a 2.50 GHz PC with 8.0 GB RAM.

### 4.1. Data Set

In this paper, we chose the three HSI data sets for experimental evaluation.

The first testing data set is Indian Pines dataset. The scene is obtained by AVIRIS sensor over the Indian Pines test site in Northwest Indiana [38]. The size of the image is 145×145 with 224 spectral reflectance bands whose wavelength ranging from 0.4 μm to 2.5 μm. Removing the crops with less coverage, we choose 9 kinds of crops in the given ground truth which are corn−notill, corn−mintill, grass−pasture, grass−trees, hay−windrowed, soybean−notill, soybean−mintill, soybean−clean and woods. Figure 1a,b illustrate the corresponding false color composition and ground truth map respectively.

The second data set is the Pavia Centre data set, which is acquired by the ROSIS sensor during a flight campaign over Pavia. The geometric resolution is 1.3 m. The image size is 1096×715×102. Due to the lack of information in the image, some samples do not contain any information. Therefore, it must be discarded before analysis. For Pavia Centre, we chose nine classes in the given ground truth: water, trees, asphalt, self−blocking bricks, bitumen, tiles, shadows, meadow and bare soil. Figure 2a,b illustrate the corresponding false color composition and ground truth map, respectively.

The third one is the Pavia University data set, also collected by the ROSIS sensor. The spatial resolution is 610×340, and it contains 103 spectral bands. The Pavia University dataset contains nine classes with the given ground truth: asphalt, meadows, gravel, trees, paninted mental sheets, bare soil, bitumen, self−blocking bricks and shadows. Figure 3a,b illustrate the corresponding false color composition and ground truth map, respectively.

### 4.2. Parameter Analysis

During the experiment, we used three evaluation indicators to measure the classification performance: OA, AA and Kappa [39]. OA represents the proportion of all correctly classified atoms to the total number of testing atoms, while AA is the average value of OAs in each class. Kappa indicates the percentage of classified testing pixels corrected by the number of agreements that would be expected by chance. Detailed definitions for each indicator can be referred to [40].

There are two main parameters (the number of adaptive dictionary atoms *K* and balancing parameter λ) that have a significant impact on classification results in our proposed PENRC. In this section, we analyze the impact of the two parameters by carrying out the sweep of the chosen parameter space and find the optimal parameters according to Figure 2. For Indian Pines, we chose 10% pixels per class as training samples. For Pavia Center, we chose 100 pixels per class as training samples and the same number for the Pavia University dataset. From Figure 4, we can see that OA increases first and then decreases with the *K* value increasing. Few adaptive dictionary atoms lack enough locality information, and too many dictionary atoms may introduce redundant category information. Then, we fixed the value of *K*, and the classification can be locally maximum with the appropriate value of λ. Then, from the maximum OA shown in Figure 4, we set *K* to 20 and λ to 1 × 10${}^{-3}$, 1 × 10${}^{-2}$ and 1 × 10${}^{-4}$ for Indian Pines, Pavia Center and Pavia University, respectively.

### 4.3. Comparisons with Other Approaches

To avoid any bias, we repeated the experiments five times and reported the average classification accuracy.

For Indian Pines, we employ 10% labeled samples per class as training set and others as testing set. The detailed partition strategy is illustrated in Table 1. From Table 2, we can see the classification performance of our proposed PENRC and J-PENRC as well as chosen compared algorithms, and the optimal results for each class are indicated in bold. For certain classes, such as grass−pasture, grass−trees, hay−windrowed and woods, the classification accuracies of our proposed PENRC and J-PENRC can be above 98%, specially for hay−windrowed, which can be up to 100%. For category soybean−clean, our algorithm improves the classification accuracy by 19.08% relative to the chosen optimal comparison algorithm ENRC. Furthermore, from Table 1, we can clearly see that our algorithms are optimal in terms of OA, AA and Kappa. In order to prove the effectiveness of our algorithm more comprehensively, we also compare the OAs, which are calculated under the different number of training samples. The classification results are shown in Figure 4. The abscissa represents the number of training samples per class, and the ordinate represents the classification accuracy. The dashed line represents OAs of the pixelwise algorithms, and the solid line represents OAs of the algorithms based on spatial information. From Figure 4, we can see that even in the case of insufficient training samples, our algorithm can achieve an ideal classification result. Furthermore, our algorithm have always been optimal compared to the same kind of contrast algorithms.

For Pavia Center, we employ 100 labeled samples per class as a training set and 2500 per class as a testing set.The detailed partition strategy is illustrated in Table 1. Table 3 illustrates the classification performance of our proposed PENRC and J-PENRC compared to other chosen algorithms, and the optimal results for each class are indicated in bold. For meadow, the classification accuracies of our proposed PENRC can be above 99.6%. For some classes, such as asphalt and tile, the classification accuracies of our J-PENRC can be above 99%. Especially for water, the classification accuracy of both PENRC and J-PENRC can be up to 100%. Furthermore, Table 3 illustrates that our proposed algorithms are optimal in terms of OA, AA and Kappa compared to other chosen algorithms. In order to further prove the effectiveness of our algorithm, we also compare the OAs of chosen algorithms under the different number of training samples. The classification results are shown in Figure 5. The number of training samples is selected from 50 samples per class to 300 samples per class. It can be seen from Figure 5 that compared with similar algorithms, our algorithm always has the best classification effect.

With regard to the Pavia University dataset, we randomly selected 100 labeled samples per class as a training set and 800 per class in the rest as a testing set (such as the shaows class, which only contains 947 labeled samples). The detailed partition strategy is illustrated in Table 1. Table 4 presents the classification result of our proposed PENRC and J-PENRC with other comparison algorithms, and the optimal results for each class are denoted in bold. For bitumen, the classification accuracy of our proposed PENRC reached 99.17%. For some classes, such as gravel and bair coil, the classification accuracy of our J-PENRC can be above 97%. Especially for meadows, painted mental sheets and shadows, the classification accuracy of J-PENRC can be up to 100%. In addition, in terms of OA, AA and Kappa, Table 3 also illustrates that our proposed algorithms are optimal compared to other chosen algorithms. In order to further prove the effectiveness of our algorithm, we also compared the OAs of the chosen algorithms with different numbers of training samples. The classification results are shown in Figure 5. The number of training samples is selected from 50 samples per class to 300 samples per class. It is evident that our algorithm always gains the most extraordinary performance.

### 4.4. Computational Complexity

In this section, we compare the computational complexity for each classifier with the Indian Pines, Pavia University and Pavia Centre datasets. All above experiments were executed five times to avoid any bias. Table 5 illustrates the total time of algorithm execution and verification of the three datasets. All experimental settings and the parameters were set to be the same as described above. As can be seen from Table 5, ENRC has a lower time complexity than PENRC. There are two reasons for this. First, ENRC uses artificial prior information to set a fixed weight parameter to combine $l_{1}$-norm and $l_{2}$-norm, while PENRC automatically learns this weight parameter through the similarity matrix. Second, the time complexity consumed by the solution approximation algorithm used by the two methods is not the same due to the difference in the math models. On the other hand, Table 5 also lists the time complexity comparison with or without LAD. Obviously, the use of LAD substantially reduces the computational complexity of PENRC and yields a better classification performance.

## 5. Conclusions

In this paper, we proposed a hyperspectral image classification algorithm named PENRC. The local constrained dictionary was first constructed to reduce the computation costs. Then, by introducing a correlation matrix, the PENRC was constructed to realize the group sparsity with self-balancing between $l_{1}$-norm and $l_{2}$-norm. The pairwise elastic net model was proven to be capable of the grouping selection of highly correlated data via establishing local, or pairwise, tradeoffs of similarity between correlation matrices, thereby rendering more robust weight coefficients. To further improve the classification performance, we also introduced spatial information and proposed the J-PENRC model. The experimental results of real hyperspectral images verified that the proposed algorithms could outperform the existing representation-based classifiers. Compared to the existing pixelwise and spatial-based algorithm, experiments on our chosen Indian Pines verified the effectiveness of our proposed PENRC and J-PENRC in quantitative and qualitative terms.

## Figures and Tables

**Figure 1 entropy-23-00956-f001:**
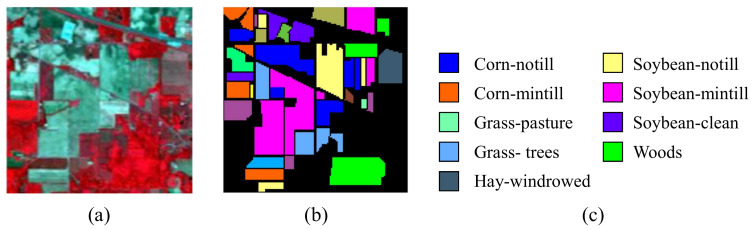
Indian Pines dataset. (**a**) composite color image. (**b**,**c**) ground truth.

**Figure 2 entropy-23-00956-f002:**
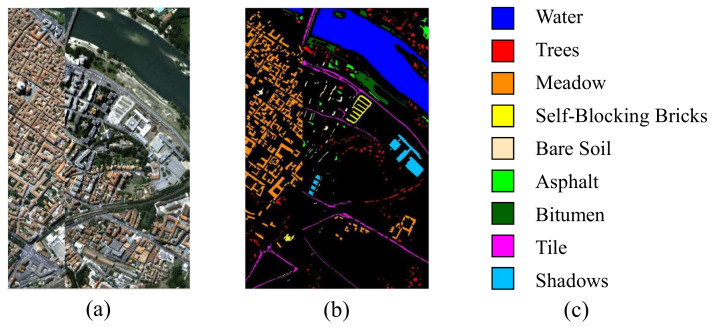
Pavia Center dataset. (**a**) Composite color image. (**b**,**c**) Ground truth.

**Figure 3 entropy-23-00956-f003:**
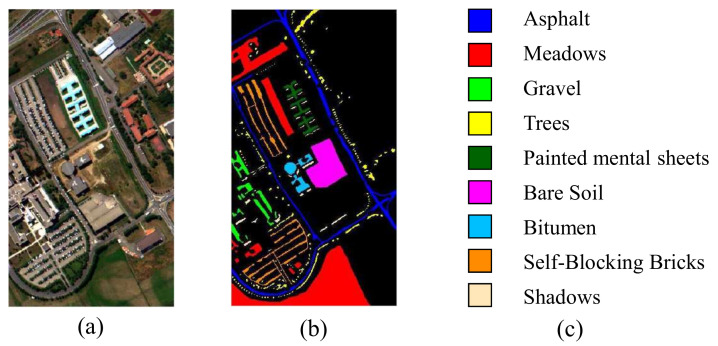
Pavia University dataset. (**a**) Ccomposite color image. (**b**,**c**) Ground truth.

**Figure 4 entropy-23-00956-f004:**
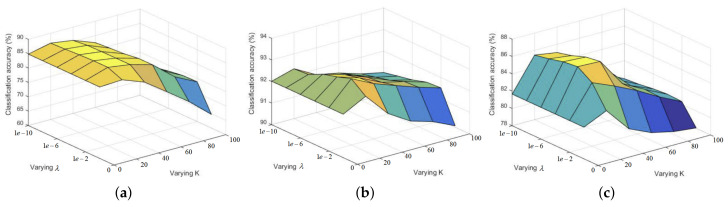
Effects of the number of adaptive dictionary atoms *K* and balancing parameter λ. (**a**) Indian Pines dataset, (**b**) Pavia Center dataset and (**c**) Pavia University dataset.

**Figure 5 entropy-23-00956-f005:**
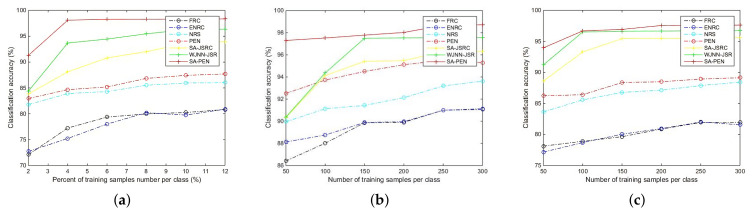
Classification performance for different numbers of training samples per class. (**a**) Indian Pines dataset, (**b**) Pavia Center dataset and (**c**) Pavia University dataset.

**Table 1 entropy-23-00956-t001:** List of the number of samples involved in training and testing for each class in Indian Pines, Pavia Center and Pavia University datasets.

	Indian Pines			Pavia Center			Pavia University		
**No.**	**Name of Class**	**Traning**	**Testing**	**Name of Class**	**Traning**	**Testing**	**Name of Class**	**Traning**	**Testing**
1	Corn-notill	142	1286	Water	100	2500	Asphalt	100	800
2	Corn-mintill	83	747	Trees	100	2500	Meadows	100	800
3	Grass-pasture	49	434	Meadow	100	2500	Gravel	100	800
4	Grass-trees	73	657	Self-Blocking Bricks	100	2500	Trees	100	800
5	Hay-windrowed	48	430	Bare Soil	100	2500	Painted mental sheets	100	800
6	Soybean-notill	98	874	Asphalt	100	2500	Bare Soil	100	800
7	Soybean-mintill	246	2209	Bitumen	100	2500	Bitumen	100	800
8	Soybean-clean	60	533	Tile	100	2500	Self-Blocking Bricks	100	800
9	Woods	127	1138	Shadows	100	2500	Shadows	100	800

**Table 2 entropy-23-00956-t002:** Classification results of Indian Pines by pixelwise algorithms (KNN, SRC, CRC, FRC, ENRC, NRS and PENRC) and spatial-based algorithms (SA-JSR, WJNN-JSR and J-PENRC). Bold indicates the best result.

No.	KNN	SRC	CRC	FRC	ENRC	NRS	PENRC	SA-JSR	WJNN-JSR	J-PENRC
1	58.44	59.48	66.49	64.03	57.69	88.99	78.44	91.60	94.16	96.75
2	54.69	62.28	63.39	64.73	67.38	89.60	78.12	86.75	92.50	97.10
3	95.00	90.38	96.54	97.96	93.09	63.78	98.46	94.01	99.77	100
4	96.70	98.97	96.70	93.91	99.46	89.96	98.98	100	100	100
5	100	98.44	99.22	99.22	98.18	98.62	100	100	100	100
6	62.68	65.33	38.10	52.87	70.16	70.30	75.62	93.59	94.17	97.90
7	79.20	78.21	90.42	90.65	82.11	69.57	90.79	95.25	95.97	97.44
8	51.72	51.72	41.38	52.66	59.60	58.45	78.68	91.18	92.32	95.92
9	93.41	94.00	98.83	95.46	96.60	98.05	99.56	97.89	98.33	99.82
**OA (%)**	75.55	76.31	78.06	80.25	79.16	86.09	87.70	94.40	96.00	98.05
**AA (%)**	76.90	77.65	77.65	79.80	80.62	80.93	88.57	94.47	96.36	98.33
Kappa	71.27	72.14	72.14	76.54	75.57	82.39	85.44	93.42	95.31	97.71

**Table 3 entropy-23-00956-t003:** Classification results of Pavia Center by pixelwise algorithms (KNN, SRC, CRC, FRC, ENRC, NRS and PENRC) and spatial-based algorithms (SA-JSR, WJNN-JSR and J-PENRC). Bold indicates the best result.

No.	KNN	SRC	CRC	FRC	ENRC	NRS	PENRC	SA-JSR	WJNN-JSR	J-PENRC
1	99.11	99.67	99.67	**100**	99.81	**100**	**100**	**100**	**100**	**100**
2	89.56	76.37	82.33	84.17	79.67	91.85	89.17	93.67	87.11	**94.67**
3	87.89	90.21	88.00	86.31	92.33	87.67	**99.67**	98.72	95.51	99.33
4	84.33	79.32	24.56	87.42	80.17	76.29	93.16	**99.82**	96.60	97.31
5	88.89	89.50	67.50	84.50	89.67	85.26	96.09	**99.00**	81.83	93.08
6	88.11	77.83	97.67	79.67	76.85	97.31	80.73	68.67	97.52	**99.41**
7	86.44	88.81	86.10	84.43	88.23	83.83	94.25	96.83	85.14	**97.28**
8	95.33	97.01	99.03	97.21	98.15	99.50	99.50	95.04	96.83	**99.60**
9	**100**	93.00	82.33	93.42	95.50	99.50	94.62	99.71	**100**	**100**
**OA(%)**	91.07	87.06	80.19	88.56	88.93	91.24	94.11	94.83	92.97	**97.78**
**AA(%)**	91.07	87.06	80.19	88.56	88.93	91.24	94.11	94.83	92.97	**97.78**
**Kappa**	89.96	86.46	78.15	87.13	87.54	90.51	93.38	94.19	92.14	**97.50**

**Table 4 entropy-23-00956-t004:** Classification results of Pavia University by pixelwise algorithms (KNN, SRC, CRC, FRC, ENRC, NRS and PENRC) and spatial-based algorithms (SA-JSR, WJNN-JSR and J-PENRC). Bold indicates the best result.

No.	KNN	SRC	CRC	FRC	ENRC	NRS	PENRC	SA-JSR	WJNN-JSR	J-PENRC
1	70.83	57.67	36.00	56.83	60.67	91.17	72.00	**94.16**	70.00	87.00
2	70.33	78.00	75.00	80.17	68.50	71.00	97.33	92.50	81.33	**100**
3	69.67	72.83	92.67	67.33	73.33	77.50	97.00	98.59	82.00	**98.67**
4	88.67	89.50	96.67	94.33	92.00	95.33	93.83	**100**	96.73	97.13
5	98.50	99.50	**100**	99.83	99.27	99.17	99.67	**100**	99.83	**100**
6	66.33	65.17	57.33	64.00	68.33	83.00	95.83	94.17	85.83	**97.00**
7	85.50	87.00	92.17	85.83	87.00	86.50	**99.17**	95.97	95.00	99.00
8	66.83	67.83	20.17	72.00	69.00	64.50	86.83	92.32	80.17	**94.33**
9	**100**	94.95	93.33	97.33	98.17	99.67	97.83	98.33	99.83	**100**
**OA(%)**	79.63	79.14	73.70	79.74	79.57	85.31	93.28	96.00	87.81	**96.69**
**AA(%)**	79.63	79.14	73.70	79.74	79.57	85.31	93.28	96.00	87.81	**96.69**
**Kappa**	77.08	76.31	70.42	77.21	77.02	83.48	92.44	95.31	86.29	**96.17**

**Table 5 entropy-23-00956-t005:** Computational complexity comparison in Indian Pines dataset.

	With/Without LAD	Running Time (s)	Overall Accuracy
ENRC	-	32.53	79.16
PENRC	✕	3472.68	85.44
	✓	72.35	87.70

## Data Availability

Not applicable.

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
