# Peer review of "Pairwise Elastic Net Representation-Based Classification for Hyperspectral Image Classification"

_entropy, 2021, doi:10.3390/e23080956_

Round 1
Reviewer 1 Report
In this paper, the authors proposed a hyperspectral image (HSI) classification method based on Pairwise Elastic Net Representation (PEN). More precisely, the local constrained dictionary has been constructed to reduce computation costs. Moreover, the pairwise elastic net representation-based classification (PENRC) is constructed by considering the correlation matrix to realize the group sparsity with self-balancing between l1-norm and l2-norm. Furthermore, spatial information has been introduced in order to improve classification performance.
For experimental results, the proposed approach has been validated on two public HSI datasets, including Indian Pines, and the Pavia Center dataset.
The proposed idea is interesting, however, some revisions have to be made and some parts of these experiments are not complete to claim the advantage of the proposed method :
1) The contributions of this paper seem to be not clear to the reviewer. Please clarify the differences between the proposed method and some existing hyperspectral image classification methods, etc. For example, which contributions are existing and which ones are your own?
2) In experimental results, do the authors have randomly selected pixels as training samples? If this is the case, what happens when you change the selected pixels (another random sampling of training data)?
3) Could the authors explain what is the motivation to use the Elastic net rather than other existing models? Could the authors add more clarifications about this?
4) The authors claimed that the proposed method outperforms the state-of-the-art HSI classification methods of HSI. How do they explain the high performance of the proposed method?
5) I suggest the authors add in the manuscript these recent references, which are related to the HSI classification based on deep learning:
- Active Deep Learning for Hyperspectral Image Classification With Uncertainty Learning, IEEE GRSL, 2021.
- Spectra-spatial Graph-based Deep Restricted Boltzmann Networks for Hyperspectral Image Classification, PIERS-Spring, 2019.
6) Moreover, I suggest the authors add in the introduction some references which are related to deep representation learning :
- Deep Spatial–Spectral Representation Learning for Hyperspectral Image Denoising, IEEE TCI, 2019.
- Mapping individual differences in cortical architecture using multi-view representation learning, IJCNN, 2020.
7) The English and format of this manuscript should be checked very carefully.
Author Response
# Reviewer 1
Comments and Suggestions for Authors
In this paper, the authors proposed a hyperspectral image (HSI) classification method based on Pairwise Elastic Net Representation (PEN). More precisely, the local constrained dictionary has been constructed to reduce computation costs. Moreover, the pairwise elastic net representation-based classification (PENRC) is constructed by considering the correlation matrix to realize the group sparsity with self-balancing between l1-norm and l2-norm. Furthermore, spatial information has been introduced in order to improve classification performance.
For experimental results, the proposed approach has been validated on two public HSI datasets, including Indian Pines, and the Pavia Center dataset.
The proposed idea is interesting, however, some revisions have to be made and some parts of these experiments are not complete to claim the advantage of the proposed method:
Reply: Thanks for the careful reading and detailed suggestions. Next, we will give our responses one by one.
1) The contributions of this paper seem to be not clear to the reviewer. Please clarify the differences between the proposed method and some existing hyperspectral image classification methods, etc. For example, which contributions are existing and which ones are your own?
Reply: As we mentioned in Section 1, the existing representation-based classification method employs the elastic net to combine the -norm and -norm together with the chosen weighting parameters. However, the optimal balance factors are all obtained by traversing the manufactured parameter space. Accordingly, in this work, we propose the pairwise elastic net representation-based classification (PENRC) method for HSI classification to automatically achieve the balance between -norm and -norm and realize the between-class sparse and in-class collaborative result. As we know, this is the first time taking advantage of the pairwise elastic net in hyperspectral image classification. Apart from this, we also adopt the local adaptive dictionary to a robust representation of dictionaries.
2) In experimental results, do the authors have randomly selected pixels as training samples? If this is the case, what happens when you change the selected pixels (another random sampling of training data)?
Reply: We do randomly select the pixels as training samples. In addition, to reduce the random selection effects, all methods are executed five times, and the mean results are reported.
Thanks for your kindly reminder, and we have made an additional clarification in the revised manuscript.
3) Could the authors explain what is the motivation to use the Elastic net rather than other existing models? Could the authors add more clarifications about this?
Reply: In the representation-based classification models, the obtained abundance coefficients reflect the importance of each training sample for reconstruction. Accordingly, the primary concern of this type of method is the solution of the abundance coefficient. For SRC, it tends to select as few as atoms. The too sparse property will lead to the deviation of final reconstruction error and the sparsity will be weakened when the number of training atoms sets is small. For CRC, it tends to select all the atoms for reconstruction, and the class discrimination will be weak when including mixed-class information. Intuitively, balance SRC and CRC to achieve better classification performance is necessary.
The Elastic Net encourages both sparsity and grouping by forming a convex combination of the CRC and SRC governed by a selectable parameter. It can be viewed as placing a global tradeoff between sparsity (-norm) and grouping (-norm). Furthermore, the Elastic Net can yield a sparse estimate with more than n non-zero weights.
In addition, the pairwise elastic net (PEN) model adopted in this paper, which using similarity measures between regressors, can establish a local balance between SRC and CRC.
Thanks for your suggestion. We have carefully revised the motivation part in the updated manuscript.
4) The authors claimed that the proposed method outperforms the state-of-the-art HSI classification methods of HSI. How do they explain the high performance of the proposed method?
Reply: The advantages of the proposed PENRC can be summarized as follows:
(1) Ideally, more corresponding non-zero coefficients are from the most relevant class. By combining -norm and -norm regularized terms together in the objective function, PENRC actually uses both advantages of CR and SR, which guarantees grouping selection on highly correlated data and enforces the intrinsic sparsity as well as self-similarity of samples simultaneously.
(2) Instead of adopting a complete dictionary of all training samples, we employ a local adaptive dictionary to obtain a more robust representation and reduce the computational cost.
(3) In addition, in order to further improve the classification performance, we also integrate the spatial information of pixel neighbourhood into the model, named joint pairwise elastic net representation-based classification (J-PENRC).
Hence, it is expected that the weight vector from PENRC reveals a more powerful discriminant ability, and thereby outperforming the original SRC and CRC.
5) I suggest the authors add in the manuscript these recent references, which are related to the HSI classification based on deep learning:
- Active Deep Learning for Hyperspectral Image Classification With Uncertainty Learning, IEEE GRSL, 2021.
- Spectra-spatial Graph-based Deep Restricted Boltzmann Networks for Hyperspectral Image Classification, PIERS-Spring, 2019.
Reply: Thanks for your suggestion. Since this study focuses more on mainstream machine learning methods, we only briefly review these references in the updated manuscript.
6) Moreover, I suggest the authors add in the introduction some references which are related to deep representation learning:
- Deep Spatial–Spectral Representation Learning for Hyperspectral Image Denoising, IEEE TCI, 2019.
- Mapping individual differences in cortical architecture using multi-view representation learning, IJCNN, 2020.
Reply: Thanks for your suggestion. We have reviewed more literature related to presentation learning in the updated manuscript.
7) The English and format of this manuscript should be checked very carefully.
Reply: Thanks for your suggestion. We have thoroughly revised and checked the language and formatting issues in the revised draft.

Reviewer 2 Report
This paper presents a hyperspectral image classification algorithm based on pairwise elastic net. The topic is interesting and to-the-point, the structure of the paper is fine, and the innovation level is fairly acceptable. The language needs to be improved though. I have two major and some more minor comments, and I believe the submitted work deserves to be published after addressing them.
Major comments:
- I suggest enriching the experimental data by adding some more challenging datasets to the two widely-used (but not that tricky!) datasets. I am pretty sure it would not be difficult to find some more standard and free data. On the other hand, most of the state-of-the-art works dealing with the same problem are using such more challenging datasets in their experiments, then it would be fair to expect the submitted work to include such data.
- I suggest adding a table (plus detailed discussions) to report the computational cost
Minor Comments:
- In Abstract, Line 4: ‘to’ is missing after ‘leading’
- In Abstract, line 7: The sentence ‘Though the similar matrix, it can realize the automatic group sparsity’ is impaired or not clear.
- In Tables, please avoid using very dark backgrounds for the table cells, since that makes it hard to read what’s written in tables when the paper is printed.
- If not necessary, please remove the line numbers for the last two lines in Algorithm 1.
- Page 4, line 116: the abbreviated form of the term ‘pairwise elastic net’ (that is PEN) is addressed here for the very first time, however, the term ‘pairwise elastic net’ has been used in the previous pages 4 times. I suggest providing the readers with the abbreviated form as soon as possible in the text.
- Conclusion is short and is just a briefed copy of the abstract.
Author Response
# Reviewer 2
Comments and Suggestions for Authors
This paper presents a hyperspectral image classification algorithm based on pairwise elastic net. The topic is interesting and to-the-point, the structure of the paper is fine, and the innovation level is fairly acceptable. The language needs to be improved though. I have two major and some more minor comments, and I believe the submitted work deserves to be published after addressing them.
Reply: Thanks for the careful reading and detailed suggestions. Next, we will give our responses one by one.
Major comments:
I suggest enriching the experimental data by adding some more challenging datasets to the two widely-used (but not that tricky!) datasets. I am pretty sure it would not be difficult to find some more standard and free data. On the other hand, most of the state-of-the-art works dealing with the same problem are using such more challenging datasets in their experiments, then it would be fair to expect the submitted work to include such data.
Reply: Thanks for your suggestion. Limited by the revision deadline, we have tried our best to enrich the experimental results. Following this suggestion, we have made an additional comparison on the Pavia University dataset in the updated manuscript.
The three public hyperspectral image classification datasets used in this work, Indian Pines, Pavia Center and Pavia University, have the highest utilization rate in recent studies. They contain two typical scenarios (rural area and urban area) in the hyperspectral applications. The abundant experimental results in this work have proved that the proposed model has advanced classification performance and great generalization ability.
I suggest adding a table (plus detailed discussions) to report the computational cost
Reply: Thanks for your advice. We have made additional analyses on computational complexity in the revised manuscript.
Minor Comments:
In Abstract, Line 4: ‘to’ is missing after ‘leading’
Reply: Thanks for your careful reading. We have checked and revised grammar issues in the updated manuscript.
In Abstract, line 7: The sentence ‘Though the similar matrix, it can realize the automatic group sparsity’ is impaired or not clear.
Reply: We have rewritten this sentence in the revised manuscript.
“This similar matrix enables automatic grouping selection of highly correlated data so that more robust weight coefficients can be estimated for better classification performance.”
In Tables, please avoid using very dark backgrounds for the table cells, since that makes it hard to read what’s written in tables when the paper is printed.
Reply: Thanks for your suggestion. Since we have illustrated the corresponding colour of each class in the previous figures, we decide to delete these colourful backgrounds in the table cells. The updated manuscript shows our adjustment.
If not necessary, please remove the line numbers for the last two lines in Algorithm 1.
Reply: We have made a corresponding modification in the updated manuscript.
Page 4, line 116: the abbreviated form of the term ‘pairwise elastic net’ (that is PEN) is addressed here for the very first time, however, the term ‘pairwise elastic net’ has been used in the previous pages 4 times. I suggest providing the readers with the abbreviated form as soon as possible in the text.
Reply: Thanks for your kind advice. We have made a revision to the updated manuscript.
Conclusion is short and is just a briefed copy of the abstract.
Reply: We have rewritten the conclusion in the revised manuscript.

Round 2
Reviewer 1 Report
The authors have revised this manuscript carefully according to my questions. I have no further questions about this manuscript. It could be accepted.
Reviewer 2 Report
The authors have completely addressed my comments and suggestions. Therefore, I believe the work can be published in its present form.
This manuscript is a resubmission of an earlier submission. The following is a list of the peer review reports and author responses from that submission.